# ENCODING THE LLM VOCABULARY BOTTLENECK

## ABSTRACT

Large Language Models (LLMs) have simplified natural language processing tasks by leveraging their ability to learn from massive volumes of data and generalise across a wide range of applications. However, as LLMs continue to scale in size and complexity, optimizing their computational efficiency has become a critical challenge. One of the major contributors to this complexity is the decision layer (referred to as the *Softmax layer*), consisting of a fully connected network matched with a Softmax activation function, which scales linearly with vocabulary size resulting in high computational costs. In this work, we first propose a framework based on Error Correcting Output Coding (ECOC), which enables the encoding of several decision boundary formulation techniques, including Softmax, to be plugged in as the decision layer of an LLM. Using this framework, we analyse the minimal design strategy for defining the simplest decision boundary with optimal computation efficiency, and propose and explore extensions to this strategy to study the accuracy-complexity trade-off compared to the Softmax-based strategy using both fine-tuning and pretraining settings. We show it is possible to maintain 90% of the Softmax accuracy when pretraining is an option, and retain 83.38% of the F1 score during fine-tuning, while using only 50% of the decision layer parameter size in both cases. Further gains arise by extending codeword length with random bits, and by increasing the intermediate hidden dimensions in MTL-ECOC. Overall, this work establishes the viability of substantially reducing the computational and architectural complexity of the output layer, while formalizing the integration of the ECOC framework within LLMs.

## 1 INTRODUCTION

Large language models (LLMs) have become the backbone of modern natural language processing (NLP), enabling breakthroughs in machine translation, summarization, code generation, and multi-turn reasoning. Their success stems from scaling transformer architectures and training on massive corpora, allowing them to model complex, long-range linguistic dependencies (Wan et al., 2024; Touvron et al., 2023). Yet, this progress comes at a steep computational and memory cost, and as their adoption grows, the efficiency challenges of LLMs have become increasingly pronounced.

Two factors are key to this challenge: (i) **sequence length**: real-world applications, such as chain-of-thought reasoning, increasingly require handling longer contexts, which expands the key–value (KV) cache linearly, straining memory bandwidth during inference; (ii) less often acknowledged is **vocabulary expansion**: supporting diverse scripts, transliterations, and multilingual use cases demands vocabularies exceeding 100k tokens (Wu et al., 2020; Bapna et al., 2022). The decision layer, implemented as a fully-connected projection with a Softmax activation, then scales linearly with vocabulary size, turning every decoding step into a compute-bound operation. While advances such as FlashAttention (Dao et al., 2022), KV-cache compression (Anagnostidis et al., 2024), Speculative decoding (Leviathan et al., 2023), LoRA (Hu et al., 2021), quantization and pruning reduce the backbone's cost, they leave this component largely untouched. Thus, even highly optimized models remain hindered by its inefficiency.

We tackle precisely this flaw: the vocabulary bottleneck of the Softmax decision layer. Our central idea is to replace Softmax with a flexible and more general *Error Correcting Output Coding* (ECOC) framework. Originating from information theory and multi-class learning, ECOC reframes multi-class classification as a collection of binary subproblems defined by a code matrix (Zor et al., 2010; 2016; Pujol et al., 2006; Bautista Martin et al., 2018; Ángel Bautista et al., 2012). Our perspective

not only reveals Softmax as a special *one-vs-all* ECOC design but also enables the encoding of alternative decision boundary formulation strategies. By adopting *minimal* ECOC designs, we use the smallest number of binary subproblems to define a multi-class decision boundary. This reduces the number of output nodes from $V$ in Softmax to just $\lceil \log_2 V \rceil$. for a problem with vocabulary of size $V$, thereby shrinking parameter counts. We further extend this with random-bit augmentations and multi-task projections, yielding robustness and accuracy trade-offs under both pretraining and fine-tuning. In brief, our contributions are:

- We introduce a principled framework that integrates ECOC as a decision layer in LLMs, with Softmax appearing as a special case.
- We design and evaluate Minimal, MinRandom, and MTL-ECOC strategies, balancing compression, robustness, and accuracy.
- We provide a thorough experimental study across pretraining and fine-tuning regimes, quantifying compression–accuracy trade-offs between parameter count, inference time, and performance.

## 2 BACKGROUND

**Parameter optimization techniques for LLMs**. Quantization and pruning form two of the most active optimisation techniques addressing backbone parameter count. Quantisation initially focused on post-training INT8 inference (Yao et al., 2022; Dettmers et al., 2021), later enabling weight and activation compression down to 4–8 bits via smoothed calibration and activation-aware scaling (Xiao et al., 2022; Lin et al., 2023). Recent pipelines such as QuaRot (Chen et al., 2024), QServe (Zhang et al., 2025), and Atom (Li et al., 2024a) demonstrate that fully low-bit inference (W4/A4/KV4) is possible with sub-point perplexity loss and significant throughput gains (Chen et al., 2024; Zhang et al., 2025; Li et al., 2024a). In parallel, pruning techniques evolved from static block removal (Wang et al., 2023b; Anil et al., 2023) to search-driven approaches like DarwinLM (Zhou et al., 2025), SlimGPT (Zhang et al., 2024), and EvoPress (Li et al., 2024b) that leverage importance scoring or second-order approximations to identify redundant substructures (Zhou et al., 2025; Zhang et al., 2024; Li et al., 2024b). Despite their efficacy, both sub-fields largely neglect the decision layer, continuing to treat the $V \times d$ matrix as a dense, unstructured tensor.

Despite significant advances in compressing the backbone of transformer models, the decision layer remains a key computational bottleneck. In decoder-only LLMs, a single forward pass over a sequence of $S$ tokens with hidden dimension $d$ and vocabulary size $V$ incurs a total complexity of

$$\mathcal{O}(LSd(d+S)) + \underbrace{\mathcal{O}(SdV)}_{\text{decision layer}}, \tag{1}$$

where the first term captures the cost of self-attention and feedforward layers, and the second term accounts for the output projection matrix mapping hidden states to logits over $V$ classes. As models scale to vocabularies of $100k$ or more tokens, the decision layer term alone can reach hundreds of millions of parameters, often surpassing the size and latency cost of the rest of the model.

Token pruning and merging methods (Bogoychev et al., 2024; Ushio et al., 2024; Bauwens & Delobelle, 2024; Lee & Hong, 2024) reduce memory footprint and decoding latency by operating directly on the input token stream, removing redundant tokens or merging similar ones, while keeping the underlying model architecture unchanged. In contrast, embedding-space techniques such as VQ-Logits (Shao et al., 2025) and CompresSAE (Kasalický et al., 2025) target the representation layer itself by compressing the vocabulary embeddings either by mapping tokens to a smaller shared codebook (VQ-Logits) or by retaining all tokens but enforcing highly sparse embeddings (CompresSAE). Importantly, both token pruning/merging and embedding-space compression methods still retain the standard dense decision layer over the vocabulary. While pruning and merging reduce sequence length and embedding-space methods optimize representation capacity, neither incorporates efficient decision boundary analysis into the design of the decision layer itself.

**Error Correcting Output Coding (ECOC)**. Given a vocabulary size of $V$, the Softmax layer produces $V$ outcomes, each of which can be interpreted as $V$ *one-vs-all* problems (see Section 3.1). When this layer is interpreted as the combination of binary sub-problems, the literature offers several options that yield better classification performance than one-vs-all (and therefore Softmax). A classic alternative is the one-vs-one voting scheme, which trains $V(V-1)/2$ pairwise classifiers and aggregates their outputs at test time (Hsu & Lin, 2002). This is where Error Correcting Output

Coding (ECOC) comes into play. ECOC offers a unified framework where the original problem can be decomposed into binary classification problems in several ways, each capturing different class groupings with their own decision boundaries.

Adopted from the communication theory by Dietterich & Bakiri (1995), ECOC is a multi-class ensemble classification strategy that decomposes a $V$-class task into $L$ binary sub-tasks governed by a code matrix $M \in \{0,1\}^{V \times L}$. Each column $j$ in the code matrix partitions the examples into a positive class, i.e. those whose class rows correspond to a 1 in that column, and a negative class, i.e. those with 0. Since each column is simply a vector of $\{0,1\}$ entries, it is often referred to as a 'bit', i.e., a binary decision unit corresponding to one node in the ensemble of classifiers; accordingly, each bit defines a two-class classification problem for training the base classifier $h_j$. Each row $M_i$ is the unique codeword for class $c_i$, where the codeword closest to the binary classifier outputs indicates the winning class. The encoding matrix can be designed according to the chosen ECOC strategy, implementing one-vs-all or one-vs-one multi-class classification approaches, as desired. For the 4-class and 5-column (i.e. 5 bit) formulation in Fig 1-(a), the five resulting binary tasks are $(+) \{c_1, c_2\}$ vs. $(-) \{c_3, c_4\}$, $(+) \{c_1, c_3\}$ vs. $(-) \{c_2, c_4\}$, $(+) \{c_1\}$ vs. $(-) \{c_2, c_3, c_4\}$, $(+) \{c_2, c_3\}$ vs. $(-) \{c_1, c_4\}$, and $(+) \{c_4\}$ vs. $(-) \{c_1, c_2, c_3\}$. Fig 1-(b) shows the corresponding decision boundaries of the individual base classifiers $h_{1,2,3,4}$.

At inference time, the ECOC ensemble produces a vector $\mathbf{y} = [y_1, \ldots, y_L]$ from base classifier outputs $y_i \in (0, 1)$. Decoding refers to the process of comparing this prediction vector with the predefined codewords to recover the most likely class. Formally, the predicted class is the one whose codeword is closest to $\mathbf{y}$: $c^{\star} = \mathrm{argmin}_{i \in \{1, \ldots, V\}} d(\mathbf{y}, M_i)$. Here, decoding can be understood as combining the set of binary decisions into a single multi-class decision, with the chosen distance metric $d$ (e.g., Hamming, Manhattan, or Euclidean) determining how the binary outputs are aggregated (Fig 1-(a)). The minimum Hamming distance between any two codewords in the formulation given in Fig 1-(a) is 2, allowing an error in any one bit to keep the correct prediction without triggering an assignment to an incorrect codeword.

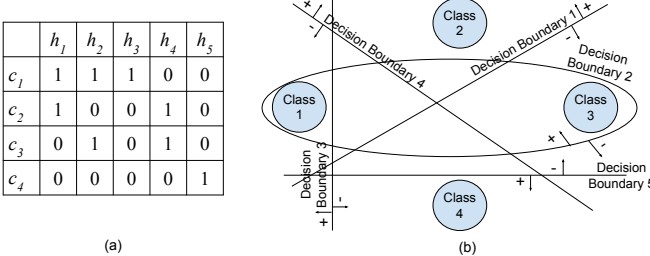

|       | $h_1$ | $h_2$ | $h_3$ | $h_4$ | $h_5$ |
|-------|-------|-------|-------|-------|-------|
| $c_1$ | 1     | 1     | 1     | 0     | 0     |
| $c_2$ | 1     | 0     | 0     | 1     | 0     |
| $c_3$ | 0     | 1     | 0     | 1     | 0     |
| $c_4$ | 0     | 0     | 0     | 0     | 1     |

(a)                    (b)

Figure 1: Example ECOC matrix and corresponding decision boundaries for a 4-class problem with $L = 5$ binary classifiers.

By simplifying the classification task into a set of loosely coupled binary problems, ECOC improves generalization, expands margins, and reduces the risk of overfitting, especially in high-cardinality output spaces (Allwein et al., 2001; Bautista et al., 2010). Examples of prior work leveraging ECOC for classification robustness for deep learning can be found in Ahmed et al. (2021); Verma & Swami (2019); Song et al. (2021); Wang et al. (2023a); Jang et al. (2025); Yu et al. (2023).

## 3 ECOC for Boundary Encoding and Optimization

In this work, we propose using ECOC to formulate a framework which facilitates various encodings of the decision layer, including those designed to minimize parameter count. Unlike prototype-tying approaches like VQ-LOGITS, this framework provides distinct representations for every token while still achieving up to logarithmic parameter scaling. The advantages obtained from optimal encodings of the decision layer can be used in tandem with embedding sparsity or quantization techniques.

### 3.1 Softmax as a One-vs-all Extension of Sigmoid

The typical decision layer based on Softmax can be interpreted as a natural extension of the sigmoid function to multi-class settings. It exponentiates each logit and normalizes by the sum of all exponentiated logits to yield a probability distribution. This implicitly brings about a set of one-vs-all comparisons where each class's score is weighed against others, resulting in the selection of the

highest probability class. During training, back-propagation updates the model by computing the cross-entropy loss for the winning class alone, reinforcing the margin between its logit and the aggregated logits of the rest. In this way, Softmax realises a one-vs-all decision rule, and fits naturally within the broader ECOC framework where one-vs-all is a special coding scheme of ECOC.

Formally, given a vector of logits $z = [z_1, z_2, \ldots, z_V]$ for a $V$ class problem, the probability assigned to class $i$ by Softmax is given by

$$p_i = \frac{e^{z_i}}{\sum_{j=1}^{N} e^{z_j}} = \frac{e^{z_i}}{e^{z_i} + \sum_{j \neq i} e^{z_j}} = \frac{1}{1 + \frac{\sum_{j \neq i} e^{z_j}}{e^{z_i}}} = \frac{1}{1 + \sum_{j \neq i} e^{z_j - z_i}} = \sigma(\Delta_i), \quad (2)$$

where $\sigma(x) = \frac{1}{1+e^{-x}}$ and the latter follows from isolating the contribution of class $i$ in the denominator and defining a logit margin term $\Delta_i := z_i - \log\left(\sum_{j \neq i} e^{z_j}\right)$. Hence $p_i = \sigma\left(z_i - \log\sum_{j \neq i} e^{z_j}\right)$ can be interpreted as a 'competition' between the logit for class $i$, $z_i$, and the 'the rest' (i.e. one-vs-all), indicated by $\log\left(\sum_{j \neq i} e^{z_j}\right)$. The bigger the gap $\Delta_i$, the closer $p_i$ is to 1. This reveals that Softmax can be expressed as a sigmoid over the margin between class $i$'s logit and the log-sum-exp of the remaining logits. Intuitively, Softmax is a soft one-vs-all decision rule: each class is compared against a smoothed aggregate of all others to define a set of binary decisions.

## 3.2 INTEGRATING ECOC IN THE DECISION LAYER

We have shown in Section 3.1 that Softmax can be framed as a one-vs-all strategy which consists of $V$ binary classifiers, each solving a *class $i$-vs-the rest* problem using the standard logistic function via sigmoid activation. This strategy can naturally be expressed using an ECOC matrix defined as an identity matrix, integrated into the decision layer of the network as shown in Fig 2-(a): The decision layer consists of a linear (fully-connected) layer that receives the decoder's hidden representation (embedding vector) and projects it onto $L$ output nodes, each corresponding to the prediction of a base classifier $h_{1,\ldots,L}$ in the ECOC matrix. These projections are passed through a sigmoid activation, and a vector consisting of $L$ probability scores is obtained. During inference, this predicted score vector is compared to a predefined binary codebook (i.e. the ECOC matrix) using a distance metric and the closest codeword is selected as the model's output, effectively combining the binary decision boundaries into a multi-class decision boundary with the chosen distance guiding the aggregation.

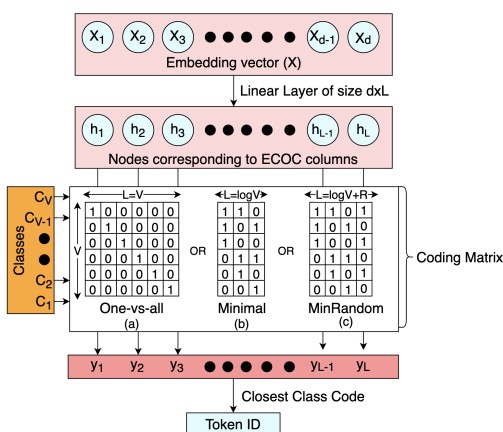

Figure 2: Decision layer in an ECOC framework. The hidden state $X$ is mapped by a linear layer into $L$ classification scores ($[y_1, y_2, \ldots, y_L]$). Three codebooks are illustrated: identity codes implementing one-vs-all, a Minimal ECOC design with $log(V)$ bits and a MinRandom design with $log(V) + R$ bits.

ECOC matrix is independent from the network backbone and can be used for formulating different multi-class decision boundary encodings. This allows the decision layer to be expressed as a modular component that maps the hidden states to an $L$-bit codeword where the choice of matrix $M \in \{0, 1\}^{V \times L}$ is determined by the ECOC strategy. Different strategies offer trade-offs between compression, robustness, and predictive performance. In this study, we mainly focus on the minimal ECOC design which employs the smallest possible set of binary classifiers to achieve maximum compression. We then expand this strategy by i) adding randomly initialised binary bits to improve robustness; and ii) assigning task-specific projection layers to each bit to enhance performance.

**Minimal ECOC**. The Minimal ECOC configuration aims to reduce the computational and memory cost of the decision layer by proposing the most compact binary encoding scheme. For a classification problem with $V$ classes (i.e., a vocabulary of size $V$), the minimum number of bits required is determined by the minimum number of distinct binary strings needed to uniquely represent each

class, which is equal to $\lceil \log_2 V \rceil$. These binary strings collectively form the Minimal ECOC matrix, where each row corresponds to the codeword of a class (Fig 2-(b)). This approach drastically reduces the number of output nodes from $V$ of Softmax (hence, one-vs-all ECOC) to $\lceil \log_2 V \rceil$, thereby compressing the decision layer and improving inference efficiency. For example, a vocabulary of 50,000 tokens would require only 16 output nodes (instead of 50,000) to represent all tokens uniquely. The minimal code length ensures that each token receives a distinct representation while introducing the smallest possible overhead.

**MinRandom ECOC**. The MinRandom ECOC extends the Minimal ECOC design by appending $R$ random bits to each codeword (Fig 2-(c)) thereby increasing its length from $\lceil \log_2 V \rceil$ to $\lceil \log_2 V \rceil + R$. This redundancy increases across-codeword distances leading to enhanced robustness and resilience to bit-level errors from individual classifiers, especially in high density vocabularies. The addition of random bits does not alter the fundamental training or decoding processes but instead expands the capacity of the decision layer, where additional flexibility helps adapt to subtle variations in the data. In our experiments (Section 4.2.2), we demonstrate that it is possible to keep the number of additional bits minimal while still obtaining significant performance gains.

**MTL-ECOC.** The multi-task learning ECOC (MTL-ECOC) decision layer builds on standard ECOC frameworks (such as Minimal and MinRandom ECOC) by introducing a task-specific transformation layer before each bit prediction. As shown in Fig. 3, each binary classifier defined by ECOC is preceded by a specific fully-connected layer, through which the embedding vector passes, allowing it to specialize in solving that binary task. This design allows each node to learn a specialized transformation, capturing finer contextual features and improving discrimination between semantically similar tokens that differ in only a few codeword bits. The additional representational capacity is advantageous in fine-tuning scenarios, where task-specific patterns are critical. Overall, MTL-ECOC combines the compactness of ECOC with the expressive power of multi-task learning, yielding higher predictive accuracy under constrained settings. We denote *Minimal MTL* and *MinRandom MTL* as the respective extensions of Minimal and MinRandom ECOC with this architecture.

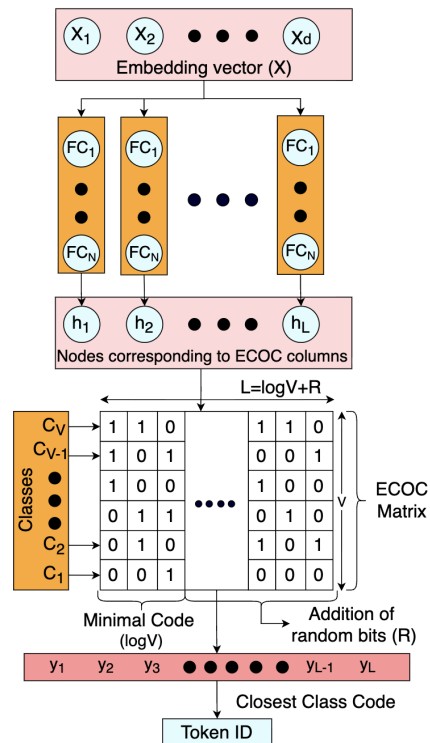

Figure 3: Relative to Minimal and MinRandom decision layers, the shared projection is replaced by lightweight, bit-specific layers, where for each bit $h_1, \ldots, h_{\lceil \log_2 V \rceil + R}$, the decoder representation $X$ passes through an independent fully-connected layer and a single-node unit to output that bit's probability.

## 4 EXPERIMENTAL EVALUATION

### 4.1 EXPERIMENTAL SETUP

To evaluate the proposed ECOC-based framework, we adopt both fine-tuning and pretraining protocols with decoder-only transformer models. For fine-tuning, we use the OPT-1.3B model (Zhang et al., 2022), a pretrained decoder architecture that balances performance and efficiency on mid-sized datasets. The ECOC decision layer is initialized with weights from $\mathcal{U}(-\sqrt{k}, \sqrt{k})$, where $k = 1/\text{in\_features}$. To enable parameter-efficient adaptation, we inject trainable LoRA adapters into the self-attention layers while keeping the backbone frozen, ensuring stability as the ECOC layer learns task-specific decoding logic. Fine-tuning is performed on the Alpaca dataset (Taori et al., 2023), a diverse instruction-following benchmark with both contextual and standalone tasks. This setup reflects realistic deployment conditions where pretrained weights remain fixed and only lightweight decision layers are trained. Hyperparameters are provided in Appendix A.1, and dataset and model details in Appendix A.2.

While practical, fine-tuning relies on representations learned during pre-training with a Softmax layer, which can bias optimization and limit ECOC's effectiveness. To isolate ECOC's capabilities, we also perform pretraining experiments, using a 15M-parameter GPT-2 (Radford et al., 2019) on TinyStories (Eldan & Li, 2023) with a 1,000-token vocabulary. Multiple ECOC configurations are tested in the decision layer to evaluate their effect on autoregressive next-token prediction. This controlled dataset enables direct comparison of ECOC and Softmax under identical conditions. Full hyperparameters and implementation details are in Appendices A.1 and A.2.

**Evaluation metrics**. Evaluation metrics are aligned with each setup. For fine-tuning, where pre-trained contextual knowledge exists, we use BERTScore to measure semantic alignment between generated outputs and ground-truth responses. Given reference $\mathbf{x} = (x_1, x_2, \ldots, x_n)$ and candidate $\hat{\mathbf{x}} = (\hat{x}_1, \hat{x}_2, \ldots, \hat{x}_m)$ sentences, BERTScore computes similarity by aligning their corresponding contextual embeddings (specifically BERT) and computing cosine similarity between them:

$$P_{\text{BERT}} = \frac{1}{|\hat{x}|} \sum_{\hat{x}_j \in \hat{x}} \max_{x_i \in x} \cos(x_i, \hat{x}_j), \ R_{\text{BERT}} = \frac{1}{|x|} \sum_{x_i \in x} \max_{\hat{x}_j \in \hat{x}} \cos(x_i, \hat{x}_j), \ F_{\text{BERT}} = 2 \frac{P_{\text{BERT}} \cdot R_{\text{BERT}}}{P_{\text{BERT}} + R_{\text{BERT}}},$$

where $\cos$ denotes the normalized cosine similarity and $P_{\text{BERT}}$, $R_{\text{BERT}}$ and $F_{\text{BERT}}$ refer to precision, recall and F-score, respectively. Unlike traditional evaluation metrics such as BLEU and ROUGE, which rely heavily on exact word matches or n-gram overlap, this metric is particularly suitable for instruction-tuned tasks requiring nuanced generation, such as paraphrasing or generalization. For pretraining, where fluency is not yet established, we use top-$k$ accuracy (with $k = 5$), which captures the frequency with which the correct token appears among the top predictions, making it well-suited to early-stage models. Together, these protocols enable us to assess both efficiency and representational quality of ECOC decision layers under distinct training regimes.

## 4.2 Accuracy Comparison of Softmax and ECOC Designs

We systematically evaluate ECOC designs with the traditional Softmax layer, emphasizing key trade-offs between accuracy, inference time, and memory complexity. Specifically, our experiments address three main questions: (1) During pretraining from scratch, how does Softmax, one-versus-all ECOC and other ECOC decision layers compare with each other on top-5 accuracy? (2) In fine-tuning, how does MTL-ECOC improve over ECOC baselines, and what steps contribute to these gains? (3) How do different ECOC designs with a fixed backbone, including longer codewords formed by adding random bits, trade off latency and memory? This analysis provides detailed insights into optimal ECOC design choices for efficient and accurate large-scale language modeling. Throughout this section, we use the notation $(x, y)$ to denote the configuration of each decision layer, where $x$ is the codeword length (i.e., the number of base classifiers) and $y$ is the intermediate hidden dimension. For instance, Minimal ECOC (10,0) indicates the minimal encoding with 10 classifiers and no MTL projection, whereas MinRandom MTL (500,128) refers to a MinRandom ECOC with 500 classifiers and an MTL projection layer of size 128.

### 4.2.1 Pretraining Setup

Table 1 reports top-5 accuracy results for the next word prediction task obtained for Softmax and several ECOC designs in the pretraining scenario using GPT2-15M (Radford et al., 2019) on the TinyStories dataset (Eldan & Li, 2023). First, following the theoretical framework from Section 2, we empirically show the link between one-vs-all ECOC and Softmax layers. The less than 0.2% difference in accuracy empirically restates that one-vs-all ECOC can represent Softmax. We then turn to the performance of the minimal design and the varying number of additional columns.

Table 1: Mean and standard deviation of top-5 accuracy across holdout test sets for different ECOC and Softmax decision layers pretrained on TinyStories using GPT2-15M as backbone model. $(x, y)$ next to each decision layer denotes the codeword length $x$ (number of base classifiers) and the intermediate hidden dimension $y$; $y = 0$ indicates no MTL projection was used.

| Decision Layer | Top-5 Accuracy (Mean ± Std) |
|---|---|
| Minimal ECOC (10, 0) | 66.82 ± 0.49% |
| MinRandom ECOC (15, 0) | 70.07 ± 0.76% |
| MinRandom ECOC (50, 0) | 75.34 ± 0.68% |
| MinRandom ECOC (500, 0) | 81.75 ± 0.53% |
| MinRandom ECOC (1000, 0) | 82.67 ± 0.41% |
| One-vs-all | 90.24 ± 0.02% |
| Softmax | 90.56 ± 0.09% |

Minimal ECOC (10, 0), with the minimal number of bits required for representing the 1000 class problem, achieves 66.82% top-5 accuracy which is about 75% of Softmax's top-5 accuracy while using only 10 output nodes, roughly 1% of the 1000 utilised by Softmax. When the minimal design is complemented with additional extra columns consisting of random bits (Section 3.2) for increased robustness, a clear upward trend in top-5 accuracy can be observed as the codeword length increases, reaching 81.75% at 500 bits. Notably, this configuration achieves 90% of the Softmax's top-5 accuracy accuracy (90.56%) while using only half the Softmax decision layer size, demonstrating the efficiency–performance trade-off enabled by ECOC. Increasing the codeword length to 1000 bits offers only marginal gains (82.67%), confirming that performance saturates around 500 bits. Note that while using the same number of bits (1000), the one-vs-all configuration performs better than MinRandom, as distinguishing one class from the rest is generally an easier task than separating randomly grouped sets of classes.

The results in this setup are obtained without MTL, as incorporating MTL in pretraining did not yield further improvements. This suggests that the model is near its representational limit, leaving little room for architectural enhancements unlike fine-tuning, which we examine in the next section.

### 4.2.2 Fine-tuning Setup

Fine-tuning experiments utilise a pretrained OPT-1.3B (Zhang et al., 2022) on the Alpaca instruction-tuning dataset (Taori et al., 2023). The top section of Table 2 reports BERTScore metrics for Softmax and several ECOC designs with Softmax achieving the strongest performance (F1 = 0.8316), given the fact model was pretrained with a Softmax layer. The goal here is to demonstrate that when pretraining is infeasible due to time or resource constraints, it is still possible to adopt a smaller decision layer design, this time by applying techniques such as additional randomisation or MTL, which enhance the representational power of the minimal ECOC design.

As a starting point, the minimal ECOC design with 16 bits reaches 50% of the Softmax performance with only 0.032% of the size (F1 = 0.4265). Aiding its design, we first expand the codebook with randomly initialized bits which improves separability, enabling MinRandom ECOC (50, 0) to achieve an F1 score of 0.4800. Second, additional representational power is obtained by introducing base classifier-specific layers through the MTL-ECOC design; Minimal MTL (16, 1024) achieves 75% of the Softmax F1 while using 67% fewer parameters. Last, when MTL and additional randomisation are combined, the gains are even greater; i.e. with MinRandom MTL (50, 1024) reaching F1 = 0.7014. Notably, for a fixed codeword length, the MTL variants consistently outperform their non-MTL counterparts. Therefore, we perform two focused ablations for this setup: the first varies the codeword length to assess how additional bits affect accuracy and compactness, and the second varies the intermediate projection dimension to measure how extra projection capacity influences precision, recall, and F1.

**Effect of codeword length**. Ablations on codeword length under MTL-ECOC design clarify how performance scales with the number of bits. As summarized in the mid section of Table 2, shorter codewords benefit most from the shared projection. Minimal MTL (16, 512) attains about 72% of the Softmax F1 while reducing the size of the decision layer by more than 80%. Increasing the bit count continues to help precision and recall, but the F1 score plateaus beyond 50 bits at 0.6934. This earlier saturation, compared with pretraining, aligns with the extra expressivity provided by the shared projection, which enables fine-grained predictions even with fewer bits.

**Effect of intermediate hidden dimension**. Ablations on the intermediate hidden dimension in the MTL-ECOC design clarify how performance scales with additional representation capacity. As shown in the bottom section of Table 2, moving from a small to a moderate dimension yields clear gains in precision, recall, and F1, while increasing the dimension further to 1024 brings only marginal improvements, indicating diminishing returns beyond a particular point. At the same dimension, MinRandom MTL consistently outperforms Minimal MTL by roughly eight to ten points in F1, with a larger lift in recall and a smaller but steady gain in precision. Further, these trends point to a saturation point at a width of 512, reserving larger widths for cases where a small extra gain justifies the additional parameters.

The above fine-tuning results describe a accuracy-complexity trade-off where complex decision layers with higher representational power result in better evaluation metrics as reported by Table 4 and can also be visualized graphically in Fig. 4 in the Appendix. The baseline model with Softmax layer

| Method | Configuration | Precision | Recall | F1 Score |
|---|---|---|---|---|
| *Softmax and ECOC Baselines* | | | | |
| Softmax | — | 0.8400 | 0.8235 | 0.8316 |
| Minimal ECOC | (16, 0) | 0.4256 | 0.4275 | 0.4265 |
| MinRandom ECOC | (50, 0) | 0.4726 | 0.4886 | 0.4800 |
| Minimal MTL | (16, 1024) | 0.6023 | 0.6047 | 0.6034 |
| MinRandom MTL | (50, 1024) | 0.6648 | 0.7439 | 0.7014 |
| *Codeword Length Ablation (MTL, size = 512)* | | | | |
| Minimal MTL | (16, 512) | 0.5924 | 0.6126 | 0.6023 |
| MinRandom MTL | (25, 512) | 0.6359 | 0.6583 | 0.6469 |
| MinRandom MTL | (50, 512) | 0.6577 | 0.7348 | 0.6934 |
| MinRandom MTL | (75, 512) | 0.6689 | 0.7278 | 0.6971 |
| MinRandom MTL | (100, 512) | 0.6835 | 0.7551 | 0.7175 |
| *Intermediate Dimension Ablation (Codeword = 16 or 50)* | | | | |
| Minimal MTL | (16, 128) | 0.5749 | 0.5925 | 0.5835 |
| Minimal MTL | (16, 512) | 0.5924 | 0.6126 | 0.6023 |
| Minimal MTL | (16, 1024) | 0.6023 | 0.6047 | 0.6034 |
| MinRandom MTL | (50, 128) | 0.6256 | 0.7078 | 0.6641 |
| MinRandom MTL | (50, 512) | 0.6577 | 0.7348 | 0.6934 |
| MinRandom MTL | (50, 1024) | 0.6648 | 0.7439 | 0.7014 |

Table 2: BERT-Score metrics on the test set for Softmax and ECOC-based decision layers. The table includes baseline results, codeword length ablation (with fixed intermediate size = 512), and intermediate dimension ablation (with codeword length = 16 or 50).

achieves the highest F1 score (0.832), but at the cost of over 100M parameters. In contrast, Minimal ECOC achieves extreme compression (33K parameters) but sacrifices accuracy (F1 = 0.427). Notably, MinRandom MTL offers a compelling middle ground, achieving competitive accuracy (F1 = 0.6934) with significantly fewer parameters (52M parameters). To complement these trends, we next examine how these decision layer choices trade time and memory complexity during fine-tuning and pretraining scenarios.

## 4.3 TIME AND MEMORY COMPLEXITY COMPARISON OF SOFTMAX AND ECOC DESIGNS

We compare different ECOC decision layers against a traditional Softmax baseline, evaluating them on three key dimensions: parameter count, training and inference time. Table 3 reports per-step

| Decision Layer | No. of Param | Acc. (%) | Frozen Backbone | | | | Trainable Backbone | | | |
|---|---|---|---|---|---|---|---|---|---|---|
| | | | Fwd | Loss | Bwd | Upd | Fwd | Loss | Bwd | Upd |
| Minimal ECOC(10,0) | 3.2k | 66.8 | 1.65 | 0.33 | 2.01 | 0.53 | 1.32 | 0.33 | 2016 | 36.6 |
| Softmax | 320k | 90.6 | 13.2 | 3.39 | 74.0 | 1.14 | 12.2 | 2.19 | 2091 | 37.2 |
| MinRandom ECOC(15,0) | 4.8k | 70.1 | 3.20 | 0.35 | 3.28 | 0.68 | 3.02 | 0.38 | 2070 | 34.7 |
| MinRandom ECOC(50,0) | 16k | 75.3 | 5.37 | 0.56 | 4.79 | 0.74 | 3.80 | 0.53 | 2025 | 38.0 |
| MinRandom ECOC(500,0) | 160k | 81.8 | 8.79 | 5.14 | 15.4 | 1.74 | 7.67 | 4.15 | 2091 | 37.2 |
| MinRandom ECOC(1000,0) | 320k | 82.7 | 13.4 | 44.5 | 80.5 | 1.94 | 11.8 | 10.5 | 2067 | 36.0 |
| One-vs-All | 320k | 90.2 | 13.0 | 44.4 | 75.5 | 1.92 | 12.0 | 10.5 | 2142 | 35.9 |

Table 3: Per-step times and accuracy for different decision layers during pretraining. All times are reported in milliseconds. Left: backbone frozen; right: end-to-end training. Abbreviations—Fwd: forward pass (through decision layer only when frozen; full model when trainable); Loss: loss computation; Bwd: backpropagation; Upd: optimizer update step.

timings and accuracy for decision layers during pretraining. The table reports four key metrics: fwd (forward pass time per batch), bwd (backward pass time per batch), loss (loss calculation time), and upd (parameter update time). Together, these measurements demonstrate how ECOC compresses the decision layer, reduces memory usage, and accelerates both training and inference compared to softmax. With a frozen backbone, the time of forward pass, loss computation, and backpropagation grows with the number of decision layer nodes $m$, while update time remains negligible. For example, loss computation time rises from 0.00033 seconds at $m = 10$ to 0.045 seconds at $m \approx 1000$, and backpropagation time from 0.0020 seconds to 0.081 seconds. This corresponds to a 20–30× speedup for Minimal designs compared to Softmax or one-vs-all, with MinRandom variants interpolating smoothly as $m$ increases. When the backbone is trainable, the backward pass of the backbone itself dominates at about two seconds, so total step time differs by only 4–7% across

decision layers, even though forward pass and loss computation times still increase with $m$. Accuracy results in Table 3 show that Minimal layers retain roughly 75% of Softmax accuracy at a fraction of computation time, underscoring the accuracy-efficiency trade-off.

| Decision Layer | Number of Parameters | Inference Time (in minutes) | F1 Score |
|---|---|---|---|
| Softmax | 102,957,056 | 12:50 | 0.8316 |
| Minimal ECOC (16, 0) | 32,768 | 01:05 | 0.4265 |
| MinRandom ECOC (50, 0) | 32,768 | 01:45 | 0.4800 |
| Minimal MTL (16, 512) | 16,785,408 | 07:00 | 0.6023 |
| MinRandom MTL (50, 512) | 52,454,400 | 08:10 | 0.6934 |

Table 4: Comparison of ECOC decision layers versus Softmax in terms of parameters, inference time and F1 score for the fine-tuning scenario. Numbers in parentheses represent codeword length and hidden-layer dimension, respectively.

Table 4 reports the inference time and the number of parameters for different decision layers in the fine-tuning scenario. The Minimal ECOC decision layer reduces over 99.9% parameters from approximately 103M to 33K, and lowers inference latency from 12m 50s to 1m 05s, an improvement of about 92%. Extending this design with multi-task learning, the Minimal MTL variant improves predictive accuracy while requiring only 16.8M parameters (an 84% reduction relative to Softmax) and an inference time of 7 minutes (a 44% reduction). The MinRandom MTL design further improves robustness and accuracy, trading additional capacity for performance while remaining more parameter-efficient than Softmax.

## 5 DISCUSSION AND CONCLUSIONS

In this work, we proposed a framework inspired by Error Correcting Output Codes (ECOC) that enables flexible implementations of the decision layer in LLMs, with Softmax appearing as a special case. Our focus was on the Minimal ECOC design and its variations, targeting computational complexity and parameter efficiency, particularly valuable in settings with extremely large vocabularies. Our experiments illustrate that ECOC-based decision layers can compress the original Softmax layer to as little as 1% of its size while retaining roughly 73.7% of baseline performance during pre-training, achieving a strong balance between efficiency and predictive accuracy. The performance improves further when randomisations are added, making it possible to achieve 90.2% of Softmax performance with 50% of the parameters. In fine-tuning, the benefits of minimal designs appear when combining additional randomisation with an MTL formulation, making it possible to obtain 83.38% of F1 using 50% of the Softmax layer size. Taken together, these results underline the practical trade-offs of adopting ECOC in LLMs. Pretraining with long codewords yields the strongest accuracy but comes with relatively higher computational cost, whereas fine-tuning frozen backbones offers more memory- and time-efficient improvements. While longer codewords and larger hidden dimensions further improve performance at the cost of time and memory, minimal ECOC designs still surpass Softmax in both efficiency and scalability, making them a strong choice for LLMs with large vocabularies.

**Limitations and Future Work**. Treating the LLM's final layer as an ECOC strategy opens access to the field's well-established ECOC best practices. We identify several promising directions for future work. A deeper understanding of the latency–complexity trade-offs, together with principled methods for determining the optimal number of bits for saturation performance (James & Hastie, 1998), remains an important step. Another avenue is the exploration of learnable ECOC variants, which can dynamically adapt coding schemes using data-driven partitioning methods. Integrating token embeddings or contextual information directly into the binary code representation may further improve semantic coherence and fine-grained predictive accuracy. Weight tying between the input embedding layer and the output ECOC projection layer also holds promise for greater parameter efficiency, stronger generalisation, and better empirical performance. At the systems level, scaling to multi-node training by distributing subsets of ECOC bits or model partitions across compute nodes could accelerate training through compartmentalisation. Finally, extending this architecture-agnostic framework to multi-modal tasks could further broaden its applicability, pushing the boundaries of ECOC's use to diverse problem domains and architectures.

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

# A APPENDIX

## A.1 HYPERPARAMETERS

| Parameter | Value | Parameter | Value |
|---|---|---|---|
| Batch size | 40 | Epochs | 3 |
| Learning rate | $2 \times 10^{-4}$ | Gradient clipping | 1.0 |
| Optimizer | AdamW | Warm-up ratio | 0.10 |
| Weight decay | 0.10 | Mixed precision | FALSE |
| Grad. accumulation steps | 1 | Group-by-length | FALSE |

Table 5: Hyper-parameters used for all fine-tuning experiments.

| Parameter | Value | Parameter | Value |
|---|---|---|---|
| Batch size | 64 | Total tokens | 1000 |
| Sequence length | 2048 | Optimiser | AdamW |
| Learning rate | $2 \times 10^{-3}$ | | |

Table 6: Pretraining hyper-parameters for GPT-2-15M on TinyStories.

## A.2 MODELS & DATASETS

**OPT Model:** The OPT (Zhang et al., 2022) family of decoder-only language models, developed by Meta AI, is designed to replicate the architecture and performance of OpenAI's GPT-3. Ranging from 125M to 175B parameters, OPT models are trained on large-scale open datasets and optimized for next-token prediction. We use the OPT-1.3B variant in our experiments, as it offers a strong trade-off between performance and memory efficiency, making it ideal for fine-tuning on mid-sized datasets.

**GPT-2 Model:** The GPT-2 model (Radford et al., 2019), is a decoder-only transformer trained with a causal language modeling objective, using next-token prediction on a large corpus of approximately 40GB of web pages filtered from Reddit links. It is available in multiple sizes: the smallest version has 124M parameters and a 768-dimensional hidden state across 12 layers, while the full model scales up to 1.5B parameters, 48 layers, and a 1600-dimensional representation. The model employs Byte-Pair Encoding (BPE) with a vocabulary of roughly 50k tokens and supports context windows up to 1024 tokens. GPT-2's versatile performance ranging from text generation and zero-shot translation and summarization made it a foundational language model and a practical choice for evaluating decision layer modifications like ECOC across various model capacities.

**Alpaca Dataset:** The Stanford Alpaca dataset (Taori et al., 2023) comprises 52,000 instruction-following examples generated via the Self-Instruct method using text-davinci-003. Each instance includes an instruction, an optional input, and a response, covering a broad range of NLP tasks. Around 40% of the examples involve contextual input, while the rest are standalone tasks such as translation or text generation. The dataset ensures diversity across tasks and is split into 85% training and 15% inference. We use Alpaca for its balanced size and task variety, enabling effective model training without the need for large-scale compute. Its accessibility and early adoption make it a strong benchmark for instruction tuning in LLMs.

**TinyStories:** The TinyStories dataset (Eldan & Li, 2023) consists of millions of synthetically generated short stories crafted by GPT-3.5 and GPT-4, using a constrained vocabulary typical of 3–4 year olds to ensure linguistic simplicity and coherence. Its primary purpose is to enable training of small language models sometimes with fewer than 10 million parameters or a single Transformer block that can still produce fluent, grammatically correct paragraphs and perform basic reasoning. In our experiments, we use a 1,000 token vocabulary variant, which provides controlled conditions for evaluating ECOC decision layers and reduces complexity during autoregressive pretraining. TinyStories offers both efficiency and interpretability, allowing us to assess the impact of compresseddecision layers on language modeling while keeping computational costs low.

### A.3    ACCURACY-COMPLEXITY TRADEOFF

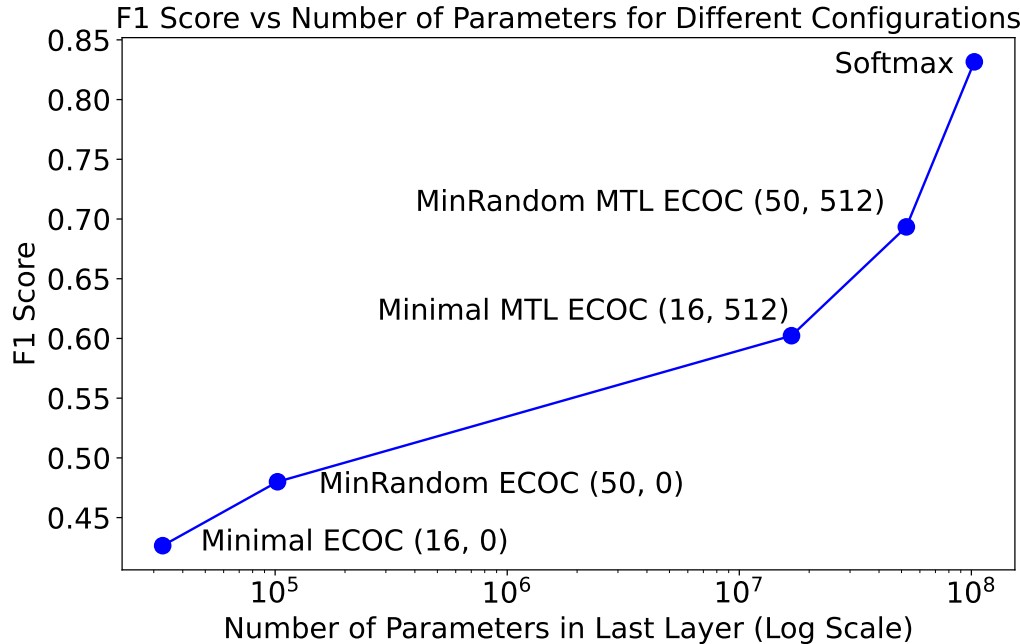

Figure 4: F1 Score vs Number of Parameters for Different ECOC decision layers and Softmax, showing the accuracy-complexity trade-off in the fine-training scenario. The x-axis is a logarithmic scale to accommodate the wide range of parameter sizes, while the y-axis represents the F1 score.

