# OpenReview forum: "Encoding the LLM Vocabulary Bottleneck"
_ICLR.cc/2026/Conference — ICLR 2026 Conference Withdrawn Submission_

### Official Review · Reviewer_4xaJ · 2025-10-25

**Soundness:** 2
**Presentation:** 2
**Contribution:** 2
**Rating:** 4
**Confidence:** 3

**Summary:**

The paper proposes replacing the LLM Softmax output layer with an Error Correcting Output Coding (ECOC) framework to mitigate the vocabulary bottleneck. It reformulates token prediction as multiple binary subproblems and introduces three variants—Minimal, MinRandom, and MTL-ECOC—to trade off accuracy and efficiency. Experiments on GPT-2-15M and OPT-1.3B show that ECOC reduces output-layer parameters and inference cost while retaining most of the performance.

**Strengths:**

1.The paper introduces an innovative ECOC-based formulation of the language model output layer.

2.The approach provides a simple yet general alternative to Softmax that substantially reduces output-layer parameters and computation, potentially benefiting large-scale LLM training and inference.

**Weaknesses:**

1.The pretraining experiment uses GPT-2-15M and the fine-tuning experiment uses OPT-1.3B, both of which are relatively small and outdated models. The paper aims to solve the “vocabulary bottleneck” problem of large LLMs, but current experiments do not convincingly show scalability to modern 7B-level  models.

2.The work is based on GPT-2/OPT while more recent backbones (e.g., LLaMA-2/3, Qwen2/3, DeepSeek) have different normalization and tokenizer designs that may affect the proposed ECOC layer. The generality claim would be stronger if tested on these.

3.The ECOC decoding does not define a proper probability distribution, so the paper reports Top-5 accuracy and BERTScore but not perplexity. It is unclear how ECOC behaves under the standard LM metric.

**Questions:**

1.Could the authors provide a more formal theoretical grounding for the proposed method? Specifically, can you analyze the error-correcting capacity or information-theoretic efficiency of the ECOC layer in the LLM setting, rather than relying purely on empirical results?

2.Have the authors compared ECOC with existing efficient-Softmax baselines such as Adaptive Softmax (Grave et al., 2017) [1] or Hierarchical Softmax (Morin & Bengio, 2005) [2]?

[1] Grave, É., Joulin, A., Cissé, M., Grangier, D., & Jégou, H. (2017). Efficient Softmax Approximation for GPUs. ICML.
[2] Morin, F., & Bengio, Y. (2005). Hierarchical Probabilistic Neural Network Language Model. AISTATS.

---

### Official Review · Reviewer_mbsP · 2025-10-27

**Soundness:** 2
**Presentation:** 2
**Contribution:** 2
**Rating:** 4
**Confidence:** 3

**Summary:**

This paper proposes a large language model output layer based on the Error Correction Output Coding (ECOC) framework, which replaces the traditional LLM output layer. The traditional LLM output layer typically consists of a linear layer and a softmax layer. The linear layer maps from the model's hidden layer dimensions to the vocabulary dimension, which is a costly multi-classification approach. The authors reduce the cost of the model output layer by decomposing the multi-classification problem into multiple binary classification problems. Assuming a vocabulary of V words, the original Softmax decision layer converts a D-dimensional vector into a V-dimensional word probability and selects words based on scores. This method converts the D-dimensional vector into an L-dimensional codeword and selects words based on the distance between this codeword and predefined codewords for the V words. Since L can be significantly smaller than V (minimum ⌈log₂(V)⌉), this approach achieves parameter compression. This work considers selection of L: minimal coding (L=⌈log₂(V)⌉) and adding random redundancy (L=⌈log₂(V)⌉+R). For vector-to-codeword conversion, it considers a DxL linear layer and a fully connected layer with an intermediate size of Y for each codeword dimension. For V=1000, using L=500 and the DxL linear layer configuration achieves 83.38% BERT-F1 and 90% top-5 accuracy at 50% of the parameter count compared to using a Softmax layer.

Furthermore, the authors conducted experiments in both the fine-tuning and pre-training stages, adjusting the balance between model capability and complexity by varying the number of redundant binary classifiers, the dimensions of the projection layer, and whether the projection layer is shared across multiple classifiers. In the pre-training stage, achieving 50% of the output layer parameters achieved 90% of the baseline performance; in the fine-tuning stage, achieving 50% of the output layer parameter size achieved 83% of the baseline performance.

**Strengths:**

(1) This work captures the inefficiency of the linear and softmax layers in the LLM output layer and proposes using the ECOC multiple binary classification framework to replace the heavy linear and softmax layers in the LLM output layer. It also theoretically proves that the softmax layer can essentially be understood as a one-to-many binary decision.
(2) Based on the extreme compression of the minimum number of binary classifiers, the authors add settings for the number of binary classifiers, the dimension of the projection layer, and whether multiple classifier projection layers are shared to adjust the balance between model capability and complexity.
(3) To more intuitively compare the complexity, the authors further calculate the time consumption of different stages of training under different configurations, such as forward inference and gradient backpropagation, to more intuitively demonstrate the efficiency of replacing the output layer with multiple binary classifications.
(4) The issues addressed in this work are significant. In practical LLM inference, the decision layer with a large vocabulary may become a major bottleneck in terms of memory consumption and computational time, making compression an important topic.
(5) In LLM inference compression, the approach of transforming direct D→V mapping into D→L and then finding the nearest neighbors of L-dimensional vectors is novel in my view and could potentially become a future research direction.

**Weaknesses:**

(1) Although the article reduces the complexity of the model output layer and provides a balance between model capability and complexity, the performance loss is still significant. When the number of parameters in the pre-training phase is half (160k), the accuracy of MinRandom ECOC(500,0) is about 90%, and when the number of parameters is equal, the accuracy is about 91%;
(2) As the model size changes, the parameter ratio of the output layer of different models may vary. The article lacks an analysis of the balance when the output layer parameter ratio is different, and an analysis of the necessity of complexity optimization when the output layer parameter ratio is different or when tied embedding is used.
(3) Figure 2 should be log2V, and Figures 2a and 2b are not found.
(4) I do not find a specific description in the paper regarding how the codewords corresponding to each word in the vocabulary are determined (i.e., the determination of the Coding Matrix). I understand that when L=V, codewords can be one-hot vectors without special design; however, when compression involves L<<V, achieving good classification performance requires the coding matrix to assign similar codewords to semantically close words, and differences in specific bits of the codeword should represent specific semantic distinctions—a task that is highly challenging (random codewords cannot achieve this). The paper should provide concrete details on how codewords are determined.
(5) For the fine-tuned model, evaluations beyond semantic similarity are lacking. The authors use BERT-F1 between generated results and ground-truth responses as the metric. While this reflects semantic accuracy, it neglects other aspects such as instruction following capability or task-specific completion ability (as described in L267). Given the specific application scenarios of current language models, supplementing performance metrics beyond semantic similarity would provide readers with greater clarity on the trade-off between compression and performance for this method.

**Questions:**

(1) The MTL term has no effect on pretraining, but does improve finetuning. Why is this so? Does less data have an effect on more models?
(2) Is there more analysis of models with different output layer ratios? How much will the balance fluctuate with different ratios? How much will the fluctuation be when the vocabulary size changes?
(3) What is the time consumption of the main model part and the output layer respectively?
(4) How is the codeword corresponding to each word in the vocabulary determined?
(5) What is the specific loss function used during training?
(6) What is the specific distance metric used during inference?

---

### Official Review · Reviewer_cCVX · 2025-10-30

**Soundness:** 3
**Presentation:** 3
**Contribution:** 3
**Rating:** 8
**Confidence:** 2

**Summary:**

This paper addresses the computational bottleneck of the Softmax output layer in Large Language Models (LLMs) by proposing a framework based on Error Correcting Output Codes (ECOC). The key contribution is reframing the multi-class prediction task as a set of binary problems, enabling drastic compression of the decision layer from $V$ output nodes to just $\lceil\log_2 V\rceil$.

Extensive experiments in both pretraining and fine-tuning regimes demonstrate that their methods can maintain 90% of Softmax's accuracy while using only 50% of the parameters, establishing ECOC as a viable and efficient alternative for the LLM vocabulary bottleneck.

**Strengths:**

This paper demonstrates notable strengths across key dimensions of scholarly work:

Originality: The work is highly original in its creative combination of the classical Error Correcting Output Codes (ECOC) framework with modern LLM architecture. While ECOC is a well-established technique in machine learning, its application to solve the vocabulary bottleneck in LLMs is novel. The key insight—framing the Softmax layer as a specific (one-vs-all) instance of a broader ECOC framework—provides a powerful new perspective for re-engineering a fundamental, yet inefficient, component of LLMs.

Quality: The paper is of high quality, evidenced by its rigorous and comprehensive experimental design. The authors validate their approach in both pre-training (GPT-2 on TinyStories) and fine-tuning (OPT on Alpaca) scenarios, providing a holistic view of the method's applicability. The ablation studies on codeword length and hidden dimensions are particularly thorough, offering clear evidence for the proposed design choices and the associated performance-complexity trade-offs.

Clarity: The paper is exceptionally clear. It builds a compelling narrative from first principles, explaining the computational bottleneck, introducing the ECOC framework, and then seamlessly showing how Softmax fits within it. The concepts of Minimal, MinRandom, and MTL-ECOC are introduced logically and supported with helpful diagrams (Fig. 2, 3). The results are presented in a well-structured, easy-to-follow manner.

Significance: The significance of this work is substantial. It tackles a critical and often overlooked bottleneck in LLM deployment: the linearly scaling output layer. By demonstrating parameter reductions of over 99% for the decision layer while retaining a significant portion of performance, the paper presents a highly practical path towards more efficient LLMs. This has direct implications for reducing the computational cost and memory footprint of LLMs, making them more accessible for resource-constrained environments. The framework is architecture-agnostic, suggesting broad applicability across model families.

**Weaknesses:**

Lack of End-to-End Latency Analysis: While the paper reports the time for the decision layer's forward pass, it omits the critical cost of the ECOC decoding step. For a vocabulary of size V, this decoding requires comparing the L-dimensional output vector against V codewords using a distance metric. This is an O(V*L) operation, which could become a new bottleneck, especially when L is large (e.g., 500). The paper should report full end-to-end inference latency (e.g., tokens/second) to prove that the parameter reduction translates into a tangible speedup in a realistic deployment scenario, rather than just shifting the computational cost.

Insufficient Comparison to Related Compression Techniques: The paper rightly notes that quantization and pruning leave the decision layer untouched. However, it does not compare against other direct approaches for tackling the output layer bottleneck.

**Questions:**

You position your method against Softmax and note that other compression techniques leave the decision layer untouched. However, a more direct alternative for reducing the output layer's cost is vocabulary pruning/trimming (e.g., Ushio et al., 2024), which directly reduces V. How does your ECOC framework, which keeps V but reduces the projection dimension, compare in terms of accuracy and efficiency to a baseline that simply uses a pruned vocabulary of size K (where the parameter count Kd is comparable to your Ld)?

---

### Official Review · Reviewer_fNCJ · 2025-10-31

**Soundness:** 4
**Presentation:** 3
**Contribution:** 1
**Rating:** 2
**Confidence:** 4

**Summary:**

The paper observes that the decision layer of LLMs (i.e., the output of softmax layer) suffers from excessive parameters and introduces ECOC to reduce the counts of parameters. In addition to proposing a substitution method, the authors also introduce two extension strategies, random binary encoding and trainable projection, to enhance the accuracy of this approach. The proposed alternative achieves 80%-90% of the softmax accuracy while reducing the model size by 50%.

**Strengths:**

1. The paper offers a novel perspective, as previous LLM optimization work has rarely focused on the output layer.
2. The theoretical analysis is relatively complete and well-structured.

**Weaknesses:**

1. The experimental validation needs to be strengthened. The paper evaluates the impact of replacing the softmax layer with ECOC during both pretraining and fine-tuning, but the fine-tuning experiments are only conducted on OPT-1.3B, and the pretraining experiments are limited to GPT-2-15M. The model scales are too small, and the experimental data are severely insufficient.
2. Even on the limited experimental data, the performance degradation caused by the proposed method is difficult to accept. In the pretraining experiments, under the most complex configuration (50,1024), the F1 score is still only 85% of softmax. In the fine-tuning experiments, with the (50,512) setting, the F1 score also remains at only about 85% of the original.

**Questions:**

1. The paper states that using random binary codewords can improve the performance of the proposed method and explains this as increased redundancy leading to higher robustness. However, this explanation seems insufficient. If random codewords can enhance performance, then structured or semantically informed embeddings might further improve it.
2. The paper does not clearly describe how the ECOC codebook is pre-generated so that each row corresponds to a token in the vocabulary.
3. The experiments are limited to GPT and OPT models. It would be beneficial to extend the proposed method to more recent models such as LLaMA or Qwen and evaluate it on additional benchmarks, such as PPL and GLUE.

---

### Note · Authors · 2025-11-21

**Comment:**

We would like to thank the reviewers for their valuable comments and feedback. We will be iterating on the article to address the comments thoroughly for a future publication.

**Withdrawal Confirmation:**

I have read and agree with the venue's withdrawal policy on behalf of myself and my co-authors.